# The Influence of Role Models on the Sedentary Behaviour Patterns of Primary School-Aged Children and Associations with Psychosocial Aspects of Health

**DOI:** 10.3390/ijerph17155345

**Published:** 2020-07-24

**Authors:** Lynda Hegarty, Marie H. Murphy, Karen Kirby, Elaine Murtagh, John Mallett, Jacqueline L. Mair

**Affiliations:** 1School of Sport, Ulster University, Shore Road, Newtownabbey, Co. Antrim BT37 0QB, UK; lynda.hegarty@nwrc.ac.uk; 2Sport and Exercise Sciences Research Institute, Ulster University, Shore Road, Newtownabbey, Co. Antrim BT37 0QB, UK; mh.murphy@ulster.ac.uk; 3School of Psychology, Ulster University, Cromore Road, Coleraine, Co. Londonderry BT52 1SA, UK; k.kirby@ulster.ac.uk (K.K.); j.mallett@ulster.ac.uk (J.M.); 4Department of Physical Education & Sport Sciences, University of Limerick, V94 T9PX Limerick, Ireland; elaine.murtagh@ul.ie; 5School of Health and Life Sciences, University of the West of Scotland, South Lanarkshire G72 0LH, UK

**Keywords:** sedentary lifestyle, mental health, physical activity, self-esteem, accelerometery

## Abstract

Background: High levels of sedentary behaviour (SB) are associated with poor health outcomes in children, but the effects on mental health are less clear. This study explored the relationship between SB and psychosocial aspects of health in children, and what influence key role models, including parents and schoolteachers, have on the SB levels of children. Methods: Physical activity (PA) and SB were measured using accelerometery in 101 children, 113 parents and 9 teachers. Children were aged 9 or 10 years old and in fourth grade. Child psychosocial outcomes were assessed using the Rosenberg Self-Esteem Scale and the Strengths and Difficulties Questionnaire. Results: Children engaged in a high volume of SB (9.6 h/day) but interrupted SB often. They accumulated less than 11,000 steps per day, and thus, many may not meet the recommended daily levels of PA. No associations were found between child SB and teacher SB during the school day or child SB and parent SB during the after-school period. No association was found between SB and self-esteem, although children with a higher body mass index had a higher number of emotional and behavioural difficulties. Conclusions: Although there was no indication that children’s SB was linked to that of parents and teachers, or that SB was associated with self-esteem or behavioural problems, school children were highly sedentary and insufficiently physically active. Therefore, there is a need to explore school practices and curriculum delivery methods, as well as school and home environments, to reduce the volume of SB children engage in.

## 1. Introduction

Large volumes of sedentary behaviour (SB) are associated with adverse health outcomes in children, such as overweight and obesity, lower fitness, lower self-esteem and prosocial behaviour, and decreased academic achievement [1,2]. High levels of screen-based SB have also been associated with more hyperactivity/inattention problems and internalising problems, and with lower wellbeing and perceived quality of life in school-aged children and adolescents [3]. Public health guidelines therefore recommend that time spent in sedentary activities should be limited in young people and adults [4].

Globally, 81% of young people (aged 11–17 years) are insufficiently physically active [5]. Research has also shown that children spend approximately 80% of their waking day sedentary [6], and that this might track from childhood through to adulthood [7], leading to an increased risk of disease. The combination of insufficient physical activity (PA) with high levels of SB creates a significant cause for concern for the future health and wellbeing of young people.

In recent years, there has been increasing attention on children’s mental health. It was reported that up to one in five children experience mental health problems [8], but the effects of SB on mental health outcomes remain relatively understudied. For example, the relationship between SB and self-esteem in children remains inconclusive [3], although there does appear to be a negative association with screen-based SB and self-esteem in children and adolescents [2,9]. From a public health perspective, it is important to understand the impact of SB on psychosocial health, particularly at key development ages, such as the period between 9–12 years when self-esteem tends to decline [10] and behavioural influences start to change. Furthermore, it is important to establish predictors and determinants of children’s SB to understand how this behaviour may be reduced.

The current evidence suggests that predictors and determinants of children’s SB remain complex and multifaceted, with further differences existing within subgroups of the child population. Behavioural influences on children’s health are affected by parents, peers and others, such as schoolteachers. Parents provide the strongest influence on children’s health beliefs [11] and the health beliefs of 8–11-year-old children are matched to that of their parents and teachers [12]. In particular, parents can influence children’s PA levels through involvement, being active role models and providing encouragement [13], but they can also influence the amount and type of SB (e.g., screen time) their children engage in [14]. School teachers are also positive role models for children within the school setting [15] and play a key role in influencing their PA behaviour [16]. Although the role of teachers in improving the health of children is secondary to that of parents [17], schools can also play a crucial role in improving the health of children [18], with teachers having the potential to act as “agents of change” [19]. Consequently, schools and teachers can play a significant role in helping to reduce sedentary behaviour in children [20].

To date, no study has explored an objective device-based measure of SB in children, their parents and the children’s schoolteachers. These data are important to establish the prevalence of SB in children and investigate the relationship between child and parent SB, and child and teacher SB. There is also little evidence on the relationship between psychosocial variables and child SB, which may provide key information on mediators and moderators of child SB. Therefore, the aims of this study were to (1) examine the SB of primary school children, and associations with parents and teacher SB; (2) compare the proportion of time spent sedentary during school versus the after-school period and (3) examine the relationship between SB and child psychosocial variables (including self-esteem and emotional state).

## 2. Materials and Methods

This study used a cross-sectional design to assess SB and correlates of SB in primary school children, their parents and teachers. Ethical approval was granted by Ulster University’s Research Ethics Committee (REC/14/0073). Participants were recruited on a rolling basis between September 2014 and March 2015 and data were collected between November 2014 and April 2015.

Permission to recruit participants was obtained from the school principals of four primary schools (urban and rural) in County Donegal (Republic of Ireland) and County Londonderry (Northern Ireland, UK). A convenience sample of 101 primary school children, 113 parents and 9 teachers were recruited for the study. These participants were derived from a possible sample of 210 children from across the four schools. Prospective participants were provided with information sheets and were given access to a short YouTube video explaining the purpose of the study. Written informed consent was obtained from teachers and parents and written informed assent was obtained from children. Sociodemographic data of children were parent-reported by questionnaire (Appendix A).

Height was measured using a portable stadiometer (Seca 220, Seca GmbH, Germany) without shoes to the nearest 0.1 cm. Weight was measured in physical education uniform and without shoes using a set of electronic weighing scales (Seca 910, Seca GmbH, Germany). Body mass index (BMI) z-scores were calculated using World Health Organization Growth Charts. Participants were classified as overweight if BMI was > 1 standard deviation and obese if BMI was > 2 standard deviations above the median.

Physical activity and SB were assessed using the activPAL^TM^ physical activity monitor (PAL Technologies Ltd., Glasgow, UK), which has been previously validated for use in child populations [21]. Participants attached the activPAL to the midpoint of the anterior aspect of the thigh using a hydrogel adhesive pad (PALstickie, PAL Technologies Ltd., UK). For extra security, a medical dressing (Mepore^®^, Molnlycke Health Care Ltd., Milton Keynes, UK) was placed directly over the activPAL onto the skin. Participants were provided with access to a YouTube video demonstrating how to apply and remove the activPALs. Participants were asked to wear the activPAL 24-h per day for seven consecutive days, removing the device only for water-based activities, e.g., showering, bathing, swimming.

The Rosenberg Self-Esteem Scale (RSES) was used to assess each child’s self-esteem [22] and is the most widely used measure of global self-esteem in the literature. The Strengths & Difficulties Questionnaire (SDQ) was used to assess emotional state [23]. Both the RSES and the SDQ were administered by the researcher and responses were self-reported by the children.

The activPAL data were downloaded using activPAL software (activPAL^TM^ Professional V5.9.1.1) and analysed using STATA software (STATA Corp LP). Quality checks and data validation procedures were carried out using an algorithm developed by Edwardson and colleagues [24]. Accelerometer data were considered valid if there were at least four days of data, including three weekdays and one weekend day [25], and a minimum of 600 min of recording between 7:00 a.m. and 10:00 p.m. Data which did not meet these criteria were excluded. Data were also examined for non-wear time whereby a 60-min non-wear time rule was applied [25]. Following quality checks, data for the entire week were exported to Microsoft Excel and categorised by activity. The activPAL data were further reduced and isolated into two specific time periods: During school (9 a.m.–3 p.m.) and after school (3–9 p.m.). This allowed two 6-h periods to be compared. Bouts of sedentary time were classified as 0–30 min and 30–60 min. Psychosocial tests were scored using the relevant protocols. Scores from the RSES were classified as low self-esteem (<15), normal (15–25) and high (26–30). Total scores from the SDQ were categorised into normal (scores of 0–15), borderline (scores of 16–19) and abnormal (scores of 20–40).

Statistical analysis was carried out using SPSS software (Version 25.0, SPSS Inc., Chicago, IL, USA). The data were visually checked for outliers using the histogram and boxplot functions and extreme data were deleted from the final data set. The Shapiro–Wilk test was used to test all variables for normality. The relationship between child, parent and teacher sitting, standing and stepping time was assessed using bivariate correlation analysis. Forced-entry multiple linear regressions and chi-square tests of independence were used to examine associations between continuous and categorical variables, respectively. Bivariate correlations were used to analyse SDQ and RSES scores against child total daily sitting time, and SDQ scores against BMI category. A two-way analysis of covariance (ANCOVA) was also conducted to examine the effect of the total scores of the RSES (three levels) and the SDQ (three levels) as independent variables on total mean sitting time (dependent variable) while controlling for BMI scores. A one-way analysis of variance (ANOVA) followed by a Tukey post-hoc test was conducted to compare accelerometer outcome measures between the children, mothers, fathers and teachers. A paired samples t-test was used to compare the children’s PA and SB during the school day with the after-school period. Results were considered significant at *p* < 0.05.

## 3. Results

Of the 101 children, 113 parents and 9 teachers recruited, 77 children (58% female), 74 parents (65% female) and 7 teachers (57% female) met valid wear time criteria for data analyses. Only five children had valid activPAL data for both parents. The remainder of the activPAL data were for one parent only. Table 1 provides the participant characteristics. The majority of the children were classified as normal weight and living with both parents.

A summary of the mean daily activity characteristics obtained from the activPAL device for the children, parents and teachers is provided in Table 2. All participants spent the majority of their waking day sedentary, with children spending the greatest amount of time, and the largest proportion of their day, sitting. There was a significant difference in total sitting time between children and mothers (*p* < 0.001), but not fathers or teachers. On average, children had a greater number of daily sit-to-stand transitions compared with mothers (*p* < 0.001), fathers (*p* < 0.001) and teachers (*p* < 0.001). Average daily step count was comparable between children, parents and teachers, ranging between 8920 and 10,142 steps per day. All four groups engaged predominantly in short bouts of sedentary behaviour (≤30 min), with very few prolonged bouts observed.

Table 3 describes PA and SB during the school day for the children and teachers and then during the after-school period for the children and parents. The majority of the children’s school day was spent sedentary (62%) while the majority of the teachers’ school day was spent standing (52%). Children spent significantly more time sitting (*p* < 0.001) and significantly less time standing (*p* < 0.001) and stepping (*p* < 0.001) compared with teachers. However, children had more sit-to-stand transitions (*p* < 0.001) and took more steps per day than teachers (*p* < 0.001).

Children engaged in similar volumes of sitting, standing and stepping during school compared to the after-school period, and there were no statistically significant differences between the two periods. However, children interrupted their sitting (sit-to-stand transitions) more during the after-school period compared to during the school day (*p* < 0.01).

During the after-school period, children spent significantly more time sitting compared to mothers (*p* < 0.001), but there was no difference in sitting time when compared with fathers (*p* = 0.130). Children recorded a higher total number of sedentary bouts (*p* < 0.001) and interrupted their sitting more frequently than parents (*p* < 0.001). Children took significantly more steps during this period compared with mothers (*p* < 0.001) and fathers (*p* < 0.01). Parental working status was used to determine differences between children’s SB during the after-school period (3–9 p.m.) and whether one, two or neither parents were at home with their children during this time. A one-way ANOVA showed no significant main effect on mean sitting time (*F* (2, 54) = 0.007, *p* = 0.993) between children who had two parents working full time (*M* = 3.52 h, *SD* = 0.71), children whose parent(s) was/were not working (*M* = 3.49 h, *SD* = 0.38) and children who had one parent at home (*M* = 3.52 h, *SD* = 0.085). For this reason, it was not considered necessary to control for parental working status in the subsequent statistical analyses.

There were no significant correlations between child and teacher mean sitting, standing and stepping time during the school day, or between child and parent mean sitting, standing and stepping time during the after-school period (Table 4). Furthermore, there were no significant correlations between child sitting time and sociodemographic variables, BMI z-scores, self-esteem or SDQ score (Table 5). However, a negative association was found between child total mean sitting time and age. A positive association was found between child BMI and SDQ scores, suggesting that participants with a higher BMI tended to report a higher number of difficulties. There was a negative association between child BMI and fathers’ total sitting time but a positive association between child BMI and mothers’ sitting time.

Scores from the RSES and the SDQ are reported in Appendix A along with classifications according to BMI category. There was no association between child self-esteem and BMI category (*p* = 0.203). A positive association existed between child SDQ score and BMI (*r* = 0.357, *p* < 0.01) and SDQ score and mother mean total sitting time (*r* = 0.421, *p* < 0.05). However, there was a moderate negative relationship between SDQ scores and fathers’ mean total sitting time (*r* = −0.509, *p* < 0.05). There was no statistically significant association between the RSES or SDQ scores on child SB when controlling for BMI category (*F* (2, 47) = 0.68, *p* = 0.935). RSES scores showed that the majority of all children scored in the ‘normal’ category. However, gender differences existed with more girls (94.1%) than boys (45.5%) scoring in the normal category. SDQ scores showed that few children scored in the ‘normal’ category, with the majority of all children scoring in either the ‘borderline’ or ‘high’ categories (Appendix A).

## 4. Discussion

To our knowledge, this study was the first to use a device-based measure of SB to explore the relationship between SB in primary school children, their parents and the children’s teachers. The main finding was that there was no association between the SB of 9–10-year-old children and that of their parents or teachers. Furthermore, sociodemographic and psychosocial variables were not related to SB levels in this age group. The data do suggest that there is a relationship between age and sitting time, suggesting that children engage in a higher volume of sitting as they get older. Furthermore, children with higher BMI scores tended to report higher scores on emotional and behavioural difficulties.

The results show that the majority of the children’s waking time was spent sitting (9.6 h/day; 63%) and that, on average, children spend more time sitting than teachers and mothers, but not fathers. This volume of SB is higher than other device-based measures of sitting time reported in the literature. For example, LeBlanc et al. [26] reported average sitting time of 8.6 h/day in 9–11-year-old children from across twelve countries worldwide, Hildebrand et al. [27] reported average total sitting time of children across five European countries to be 6.2 h/day^−1^ and Yildirim et al. [28] reported total sitting time of time in boys and girls across five European countries to be 7.5–8.7 h/day and 7.6–8.5 h/day, respectively. Our data suggest that children in this sample are more sedentary than their counterparts in other countries. The proportion of time spent sitting during the school day was also high in the current sample (62%) but comparable to data from other European countries, including Belgium (63%), Greece (63%), Hungary (67%), Netherlands (66%) and Switzerland (63%) [29]. This suggests that activities outside of school may have a greater influence on children’s overall SB than activities during school. For example, passive transportation (including car and bus travel) is the predominant school commuting method in Ireland [30], while active commuting to school (including. walking and cycling) is more common in other countries, such as Sweden and Finland [31,32]. It should be noted that the data from this study were collected during the winter months, when opportunities for outdoor play and active travel to school are lessened due to shortened periods of daylight and poor weather conditions. Previous research suggests that children tend to be less active during winter months due to inclement weather [33]. More research is needed to clearly understand patterns and types of PA and SB accumulation by primary school-aged children in different domains and taking into account seasonal variation. Despite high levels of SB, both children and parents in this study accumulated around 10,000 steps per day. Whereas this indicates that parents were achieving approximately 30 min of moderate-to-vigorous PA and were therefore physically active [34], children fell short of the estimated step count associated with achieving at least 60 min of moderate-to-vigorous PA per day [35]. Although we cannot confirm the intensity at which these steps were accumulated, it is possible that the PA level observed in parents might offset the health risks associated with high volumes of SB [36], but it appears that children are not active enough to counteract these health risks.

Children in the current study engaged in a higher number of interruptions to SB (sit-to-stand transitions) during the after-school period compared to during the school day. This suggests that children naturally break up periods of sitting more when out of school compared to during school. This finding supports that of Kwon and colleagues [37] who found that, during school hours, children had fewer breaks in SB than any other time of the week. Within the school setting, there is a general expectation by teachers that children should ‘sit still’ [38] which, based on our data, appears to go against the children’s natural instinct to interrupt their SB. It is possible that this expectation actually contributes to or influences a child’s habit formation toward more sitting. Additionally, there was no relationship between child and teacher SB during the school day. Given a traditional education system in which children are expected to sit during lessons while the teacher stands to teach, the lack of relationship may not be surprising. In light of these findings, school-based interventions aimed at interrupting and reducing children’s SB should consider teaching culture, curriculum delivery methods and how SB breaks can be incorporated into daily teaching and learning activities. Changes in pedagogical approaches to curriculum delivery have the potential to modify children’s sitting time [20]. Recent examples include physically active learning, such as exercising while spelling or doing math, using a jump mat to answer questions, going in a virtual active field trip and classroom movement breaks, such as running and jumping on the spot [39,40]. Both physically active learning and classroom movement breaks have been shown to increase PA [39], specifically by displacing SB with either light PA or moderate-to-vigorous PA [40]. However, our data suggest that encouraging teachers to reduce their SB in order to reduce the children’s SB in class may be an ineffective approach.

Investigating the synchronous patterns of SB by all members of a family unit may provide useful insight to the social determinants of children’s SB. Findings from the current study suggest that the SB of the parent has little impact on child SB, as there was no association between child and mother or father sitting time during the after-school period. This contrasts slightly with previous research. For example, Jago and colleagues [41] previously found an association between mother and daughter total sitting time but not mother and son total sitting time during a five-day period including two weekend days. However, the study did not account for behaviours in different settings, i.e., school versus home, nor did it explore associations with fathers. Hughes and colleagues [42] found a correlation between child sitting time on weekend days but not weekdays. It should be noted that the analysis methods employed in these studies differ from the process followed in the current study, which used data from at least three weekdays and one weekend day. It is possible that there are distinctions between behaviours that occur during the week when children are at school and during weekends when families are more likely to spend time together. The lack of associations in after-school SB between children and parents in the current study may be explained by different activities, such as hobbies and homework, which do not require the involvement of the parent, and domestic chores, which may not involve children. Although we did not assess activity type in this study, mothers had greater standing time than the children and fathers, which may be explained by higher levels of domestic activity [43]. Consequently, the SB of children during the after-school period appears be largely uninfluenced by the SB of the parents, but it is possible that a different relationship exists during the weekend. Siblings also have a meaningful influence on a variety of PA outcomes [44], therefore it is possible that siblings have a greater effect on SB than parents do. Future research should examine the influence of siblings as agents of change in relation to SB.

The data from this study indicate that child total SB is not related to self-esteem. Nihill and colleagues [45] also found no relationship between objectively measured SB and self-esteem in adolescent girls but did report a negative association for computer use and watching DVDs. Others have also reported a negative relationship between screen-based SB and self-esteem [1,2]. A systematic review of SB and health indicators in children reported that children watching TV for more than 2 h per day was associated with lower self-esteem [2]. Time spent in screen-based SB appears to have greater influence on mental health outcomes, but longitudinal evidence is needed to confirm this.

There was no relationship between child SB and SDQ scores in the current study, but the data did show a positive relationship between SDQ score and BMI category, suggesting that children classified as overweight or obese reported more emotional and behavioural difficulties. The children’s mean total scores for the SDQ were higher than those reported elsewhere for children of a similar age [46]. Indeed, the analyses showed that majority of participants fell into the ‘borderline’ category (44.6%), regardless of gender, and an ‘abnormal’ SDQ total score was found in 32.4% of participants, suggesting a higher risk of mental health problems in this sample.

The strengths of this study include the device-based measurement of SB and comprehensive description of SB during the school day and after-school period, as well as the examination of factors influencing SB in primary school children, including the SB of parents and teachers. As such, the study makes a significant contribution to the growing body of evidence on child SB. The limitations of the study include the cross-sectional design, which prevents an understanding of causality. Although our initial sample size was powered to detect between group differences, almost 30% of accelerometer data were deemed invalid by the adopted wear time protocol. Given the resultant sample sizes for children (*n* = 77), mothers (*n* = 48) and fathers (*n* = 26) providing valid accelerometer data, our findings may be underpowered, and therefore comparisons between groups should be treated with caution. In order to achieve 80% statistical power on the various tests of group differences, the minimum detectable effect sizes should be *d* = *0*.52, 0.65 and 0.70 for tests comparing children with mothers, children with fathers and mothers with fathers, respectively. Furthermore, the small teacher sample size may have been insufficient to identify relationships with child behaviour. The 10-h wear time protocol used to obtain valid data was based more on common practice than evidence. Therefore, given possible differences in waking time between the children and adults, different wear time criteria may have yielded more valid data for analysis. In addition, while device-based measurements are considered more accurate than subjective methods in quantifying PA and SB [47], it is not possible to distinguish activity type or contextual information about behaviour. As such, it was not possible to control for types activity when comparing SB during and after school. Finally, data were collected between November to April. Thus, it is also possible that seasonal variations altered SB, particularly in relation to the after-school period [48].

## 5. Conclusions

The results of this study highlight the large portion and high volume of time spent sedentary by 9–10 year-old primary school children. There is a need to further explore school practices, curriculum delivery methods, policies and environments to reduce SB, as well as ways in which parents and other family members can encourage less sitting during after-school periods. The study found no relationship between child and teacher SB during the school day and no relationship between child and parent SB during the after-school period, suggesting that teacher and parent SB have little influence on child SB. Furthermore, the total SB of children appears to be unrelated to mental health indicators, such as self-esteem. The results highlight the complex nature of the influences on child SB and that predictors may be a combination of a wide range of variables rather than a definitive few. Future research should look to compare the influence of role models on SB in other age groups, particularly at transitional phases in childhood and adolescence.

## Figures and Tables

**Table 1 ijerph-17-05345-t001:** Child participant characteristics.

	All (*n* = 77)	Boys (*n* = 32)	Girls (*n* = 45)
Age (years)	10.2 (0.4)	10.2 (0.4)	10.2 (0.4)
Height (cm)	140.2 (6.1)	140.8 (6.2)	139.8 (6.0)
Body Mass (kg)	37.0 (9.9)	26.8 (9.9)	37.1 (9.9)
BMI Category (%)			
Thin	12.2	12.9	11.6
Normal	48.6	54.8	44.2
Overweight	24.3	19.4	27.9
Obese	14.9	12.9	16.3

Values are means (standard deviation; SD) or percentages.

**Table 2 ijerph-17-05345-t002:** Total daily physical activity and sedentary behaviour.

	Child (*n* = 77)	Mother (*n* = 48)	Father (*n* = 26)	Teacher (*n* = 7)
Sitting time (h)	9.6 (1.9)	8.4 (1.7) *	9.1 (1.6)	8.6 (1.2)
Standing time (h)	3.4 (1.0)	5.4 (1.6) *	4.5 (1.4) *	6.0 (1.0) *
Stepping time (h)	2.2 (0.6)	2.0 (0.6)	2.1 (0.7)	2.0 (0.5)
Sitting time (%)	63	53	58	52
Standing time (%)	22	34	29	36
Stepping time (%)	15	13	13	12
Sit-to-stand transitions (*n*)	87 (16)	58 (14) *	65 (23) *	37 (16) *
Sedentary bouts 0–30 min (*n*)	83 (17)	55 (15)	61 (24)	52 (23)
Sedentary bouts 30–60 min (*n*)	3 (1)	3 (1)	3 (1)	2 (1)
Average daily steps (*n*)	10,142 (3016)	9450 (3230)	10,010 (3422)	8920 (2694)

Values are means (SD); Percent values represent the percentage of activPAL wear time for each outcome. * indicates a significant difference (*p* < 0.05) compared with children.

**Table 3 ijerph-17-05345-t003:** Physical activity and sedentary behaviour during and after school.

	School Day 9.00 a.m.–3.00 p.m.		After School 3.00 p.m.–9.00 p.m.		
	Child (*n* = 77)	Teacher (*n* = 7)	Child (*n* = 77)	Father (*n* = 26)	Mother (*n* = 48)
Sitting time (h)	3.6 (0.8)	2.1 (0.4) *	3.5 (0.7)	3.3 (0.7)	3.0 (0.7) *
Standing time (h)	1.3 (0.6)	3.0 (0.5) *	1.3 (0.4)	1.5 (0.5)	2.0 (0.6)
Stepping time (h)	0.9 (0.3)	0.7 (0.1) *	1.0 (0.4)	0.7 (0.4)	0.8 (0.3)
Sitting time (%)	62	36	60	60	52
Standing time (%)	22	52	23	27	34
Stepping time (%)	16	12	17	13	14
Sit-to-stand transitions (*n*)	33 (12)	20 (7)*	38 (9) ^†^	23 (12)	24 (9)
Sedentary bouts 0–30 min (*n*)	32 (12)	20 (7)	37 (10)	22 (12)	23 (9)
Sedentary bouts 30–60 min (*n*)	1 (1)	0 (0)	1 (1)	1 (1)	1 (1)
Average daily steps (*n*)	4486 (1366)	3226 (590) *	4476 (2062)	3256 (1814) *	3528 (1522) *

Values are means (SD); * indicates (*p* < 0.01) compared with children; ^†^ indicates a significant difference (*p* < 0.01) compared with the school day.

**Table 4 ijerph-17-05345-t004:** Sedentary behaviour associations between participant groups.

	Variable	Pearson’s *r*	Significance
During School	*Child versus teacher*		
	Sitting time	−0.01	0.919
	Standing time	0.04	0.753
	Stepping time	0.10	0.476
After School	*Child versus mother*		
	Sitting time	−0.04	0.799
	Standing time	−0.13	0.386
	Stepping time	0.17	0.239
	*Child versus father*		
	Sitting time	0.30	0.143
	Standing time	0.14	0.481
	Stepping time	0.26	0.198

**Table 5 ijerph-17-05345-t005:** Associations between child sitting time, sociodemographic and psychosocial variables.

	Variable	Pearson’s *r*	Significance
Total Child Sitting Time	Age	−0.31	0.016 *
	Body Mass of child	−0.15	0.262
	BMI z-score	−0.23	0.085
	SDQ	−0.24	0.076
	RSES	−0.04	0.761
After School Sitting Time	Structure of the child’s family unit	0.18	0.124
	Father’s highest level of education	−0.13	0.278
	Mother’s highest level of education	−0.14	0.240
	Child’s race/ethnicity	0.17	0.157
	Work status of father	0.03	0.826
	Work status of mother	−0.14	0.250
	No. of people living in house	−0.20	0.095
	Household income	−0.01	0.945
BMI z-scores	SDQ	0.36	0.002 **
	RSES	−0.10	0.400
SDQ	RSES	−0.00	0.989
	Mother mean total sedentary time	0.42	0.026 *
	Father mean total sedentary time	−0.51	0.037 *
RSES	Mother mean total sedentary time	−0.14	0.486
	Father mean total sedentary time	0.41	0.088

BMI = Body Mass Index; RSES = Rosenberg Self-Esteem Scale; SDQ = Strengths and Difficulties Questionnaire; * *p* < 0.05 level (2-tailed); ** *p* < 0.01 level (2-tailed).

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
