# Peer review of "The Influence of Role Models on the Sedentary Behaviour Patterns of Primary School-Aged Children and Associations with Psychosocial Aspects of Health"

_ijerph, 2020, doi:10.3390/ijerph17155345_

Round 1

Reviewer 1 Report

Overall Comments

This is an interesting and well-written paper that examine the influence of key adults on children’s sedentary behavior.  The findings of this study are interesting, but there are key pieces of information that are not included that would make the results stronger.

Specific Comments

Abstract

Please indicate the age of the children and the grades of the children and teachers.

Methods

In the analysis, consider controlling for factors that can influence your outcomes such as the school, area of the country, and weather (winter/spring). Also consider performing separate analyses for weekday versus weekend day.

Results

The environment and external influences has a large influence on sedentary behavior- weather they are working, at home, or at school. Consider controlling for weather the parent is working outside the home.

Lines 151-52.  Does the study have sufficient power to look at differences between mothers/fathers and their children? Teachers and their students?

Line 168-72: The after school period is a time where it can be scheduled or unscheduled, allowing the child the most control over their behavior (or not).  Please describe what activities were performed during the after school period (sport, another activity, etc.).  Further, consider controlling for after school activity or sport.  If the students were at home, it is important to understand whether the parent was also at home, or whether they were working (where they would have less control over their SB).  Please include this data as well.  Finally, it would be interesting to understand the relationship between parent-child SB on a weekday vs weekend, where the families may have more control over their behavior and spend more time together.

Reviewer 2 Report

Thank you for your well written manuscript.

The paper sets up a rationale for an exploration of curriculum delivery methods. However the paper contains little detail about what this could entail or how this could be achieved in an already crowded curriculum.

Whilst acknowledging the data collection period i think this point needs further expansion. Is this a global trend with winter , colder temperatures and shorter periods of daylight.

Did the research team collect teachers age ? 

It is worth noting that future studies need to also take into account teachers views in what a traditional education system looks like. Hopefully teachers are becoming increasingly aware of the benefits of active learning this breaking sitting periods.It would be beneficial to include example here of approaches to reduce sedentary time particularly through active lessons, energizers etc to enhance curriculum learning.

A very clear well written  paper with potential for future research to support. A comparison with other age groups would be beneficial. 

Round 2

Reviewer 1 Report

The authors have addressed the comments in a sufficient manner.